# Optimization of High-Density Fermentation Conditions for *Saccharomycopsis fibuligera* Y1402 through Response Surface Analysis

**DOI:** 10.3390/foods13101546

**Published:** 2024-05-16

**Authors:** Hongyang Yuan, Qi Sun, Lanshuang Wang, Zhilei Fu, Tianze Zhou, Jinghao Ma, Xiaoyan Liu, Guangsen Fan, Chao Teng

**Affiliations:** 1Key Laboratory of Geriatric Nutrition and Health (Beijing Technology and Business University), Ministry of Education, Beijing 100048, China; enzyme513@163.com (H.Y.); s17321914603l@163.com (Q.S.); lanshuangwang@outlook.com (L.W.); ztz18562536917@163.com (T.Z.); 15032049018@163.com (J.M.); tc2076paper@163.com (C.T.); 2School of Biology and Food Science, Hebei Normal University for Nationalities, Chengde 067000, China; 13833415159@163.com; 3China Food Flavor and Nutrition Health Innovation Center, School of Food and Health, Beijing Technology and Business University, Beijing 100048, China; 4Sweet Code Nutrition & Health Institute, Zibo 256306, China

**Keywords:** high-density fermentation, response surface analysis, *Saccharomycopsis fibuligera*, total colony count

## Abstract

*Saccharomycopsis fibuligera*, which produces enzymes like amylase and protease as well as flavor substances like β-phenyl ethanol and phenyl acetate, plays a crucial role in traditional fermented foods. However, this strain still lacks a high-density fermentation culture, which has had an impact on the strain’s industrial application process. Therefore, this study investigated the optimization of medium ingredients and fermentation conditions for high-density fermentation of *S. fibuligera* Y1402 through single-factor design, Plackett–Burman design, steepest ascent test, and response surface analysis. The study found that glucose at 360.61 g/L, peptone at 50 g/L, yeast extract at 14.65 g/L, KH_2_PO_4_ at 5.49 g/L, MgSO_4_ at 0.40 g/L, and CuSO_4_ at 0.01 g/L were the best medium ingredients for *S. fibuligera* Y1402. Under these conditions, after three days of fermentation, the total colony count reached 1.79 × 10^8^ CFU/mL. The optimal fermentation conditions were determined to be an initial pH of 6.0, an inoculum size of 1.10%, a liquid volume of 12.5 mL/250 mL, a rotation speed of 120 r/min, a fermentation temperature of 21 °C and a fermentation time of 53.50 h. When fermentation was conducted using the optimized medium and conditions, the total colony count achieved a remarkable value of 5.50 × 10^9^ CFU/mL, exhibiting a substantial increase of nearly 31 times the original value in the optimal culture medium. This significant advancement offers valuable insights and a reference for the industrial-scale production of *S. fibuligera*.

## 1. Introduction

*Saccharomycopsis fibuligera* is a non-brewing yeast that can produce amylase, acid protease, β-glucosidase, and other important enzyme systems indispensable in fermentation [1,2,3,4,5]. Therefore, this yeast can use carbohydrates such as sucrose, cellobiose and starch in the fermentation substrate to produce ethanol and various flavor substances. In food fermentation, it plays an essential role and is a key yeast for the synthesis of esters and aromas [6,7,8]. For example, *S. fibuligera* KJJ81 can produce volatile flavor substances such as β-phenylethanol, phenyl acetate and ethyl phenylacetate in YPD medium [6]. Furthermore, it was discovered that *S. fibuligera* could be identified in a greater percentage of fermented foods during the process of meticulous analysis of the microbial population in fermented foods. This result indirectly indicates the significance of *S. fibuligera* in the formation of several fermented foods [9,10,11,12,13,14,15,16].

Considering the significance of *S. fibuligera* in fermented foods, several researchers have undertaken extensive investigations on the internal processes of this strain [17,18,19]. The constant acquisition and mastery of relevant knowledge about this strain facilitate its industrial production, improving its use in fermented foods. To achieve industrial production of this strain, it is first necessary to achieve high-density cultivation and fermentation of the strain, which is a crucial step for the better application of this strain in fermented foods [20]. However, although it is a promising microbial strain for food fermentation, there is a lack of research on the high-density fermentation of *S. fibuligera* Y1402. Consequently, this yeast requires immediate attention from researchers interested in studying high-density fermentation techniques.

Research on high-density fermentation in different strains indicates that the most important factors that affect the high-density growth of yeast cells in batch fermentation are the ingredients in the medium and the conditions of fermentation [20,21,22,23,24]. Research has shown that glucose, sucrose, yeast extract, peptone, KH_2_PO_4_, MgSO_4_, and CuSO_4_ are the main nutrients that promote yeast cell growth; at the same time, the pH of the medium, inoculum size, rotation speed, liquid volume, fermentation temperature and fermentation time are the key fermentation conditions affecting yeast cell growth [23]. Zhang et al. [20] conducted single-factor design (SFD) combined with response surface analysis (RSA) to optimize the carbon source, nitrogen source, and inorganic salt components in the medium. With the optimized conditions, the viable cell count of *Hanseniaspora uvarum* BF-345 reached 5.67 × 10^9^ CFU/mL. Sun et al. [21] used SFD, Plackett–Burman design (PBD), steepest ascent test (SAT), and central composite design to determine the optimal ingredients in the medium and the best fermentation conditions for *Lactiplantibacillus plantarum* NCU137. After optimization, the viable cell count reached 9.20 × 10^9^ CFU/mL. Due to the different characteristics of different strains, there are significant differences in the optimal growth nutrients and fermentation conditions. Therefore, it is necessary to optimize the medium ingredients and the fermentation conditions for each strain individually [20].

In our previous work, we isolated a yeast strain of *S. fibuligera* Y1402 from the environment of Baijiu (Chinese liquor) production. This strain demonstrated high production of flavor compounds such as 3-methylthiopropanol and phenylethanol [25]. It significantly improved the quality of tobacco products when applied to tobacco leaf fermentation (related research is still pending publication). Therefore, it holds great potential for practical applications. This study aims to optimize the medium ingredients and the fermentation conditions for *S. fibuligera* Y1402 using SFD, PBD, SAT, and RSA. This optimization will help us determine the optimal high-density fermentation conditions for this strain. Subsequently, we will validate the results through scaled-up fermentation in a 3 L fermenter, providing theoretical insights for the industrial production of *S. fibuligera*.

## 2. Materials and Methods

### 2.1. Strain and Reagents

*S. fibuligera* Y1402, used in this study, was collected from the Baijiu brewing environment and preserved in our laboratory. All other chemicals were analytical grade and commercially available unless otherwise stated.

### 2.2. Medium Preparation

Yeast extract peptone dextrose (YPD) medium: yeast extract powder 10 g/L, peptone 20 g/L, and glucose 20 g/L, pH 5.8.

The detailed recipes for basic media 1 to 9 used for the high-density culture of *S. fibuligera* Y1402 are shown in Table 1. All media were autoclaved at 115 °C for 20 min.

### 2.3. Yeast Activation and Basic Medium Selection

The preserved *S. fibuligera* Y1402 was inoculated in a 250 mL Erlenmeyer flask containing 50 mL YPD medium. The flask was incubated on a shaker at 30 °C and 180 rpm for 24 h. The activated yeast was again transferred to 50 mL of YPD medium, with an inoculum size of 0.5%.

The activated *S. fibuligera* Y1402 fluid was inoculated into 9 types of basic medium (50 mL/250 mL) with an inoculum size of 1%. Fermentation was carried out at 180 r/min and 30 °C for 3 days. Cell proliferation was measured by assessing cell suspensions’ turbidity (optical density at 560 nm) using a spectrophotometer (Beijing Puxi General Instrument Co., Ltd., Beijing, China), following the methodology specified in a prior study [26]. The OD_560nm_ value was measured to select the optimal basic medium.

### 2.4. Optimization of Medium Ingredients

#### 2.4.1. Single-Factor Design of Medium Ingredients

Concentrations of various ingredients, including glucose, yeast extract, peptone, KH_2_PO_4_, MgSO_4_, and CuSO_4_, were optimized for *S. fibuligera* Y1402 production by SFD (Table 2).

#### 2.4.2. Plackett–Burman Design of Medium Ingredients

According to the results from the SFD, six variables, including glucose, peptone, yeast extract, KH_2_PO_4_, MgSO_4_, and CuSO_4_, were selected for PBD at two levels denoted high (+1) and low (−1) (Appendix A). PBD was used to design a set of N = 15 experimental combinations (Table 3) in order to investigate the significance of the effects of each factor on the growth of *S. fibuligera* Y1402.

#### 2.4.3. Steepest Ascent Test of Medium Ingredients

The SAT was constructed with three important variables (glucose, yeast extract, and KH_2_PO_4_) obtained from the PBD. The direction of each variable in the SAT was determined according to the regression results of the PBD, and the step length of each variable was determined by the regression model from the SFD, PBD, and practical experience (Table 4).

#### 2.4.4. Response Surface Analysis of Medium Ingredients

After the center point was identified using the SAT, RSA was used to optimize the first three variables [glucose concentration (*X*_1_), yeast extract concentration (*X*_2_), KH_2_PO_4_ concentration (*X*_3_)] screened according to their statistically significant influence on *S. fibuligera* Y1402 production by PBD. Seventeen test groups with three factors and three levels were designed based on the Box–Behnken experimental design (Table 5).

### 2.5. Optimization of Fermentation Conditions

#### 2.5.1. Single-Factor Design of Fermentation Conditions

After the optimization of medium ingredients, various fermentation conditions, including initial pH, inoculum size, liquid volume, rotation speed, fermentation temperature and fermentation time were optimized using SFD (Table 2).

#### 2.5.2. Plackett–Burman Design of Fermentation Conditions

According to the results from the SFD, five factors including initial pH, inoculum size, rotation speed, fermentation temperature and fermentation time were selected for PBD at two levels denoted high (+1) and low (−1) (Appendix A). The PBD was used to design a set of N = 15 experimental combinations (Table 6).

#### 2.5.3. Steepest Ascent Test of Fermentation Conditions

The SAT was constructed with three important variables (inoculum size, fermentation temperature and fermentation time) obtained from the PBD (Table 7).

#### 2.5.4. Response Surface Analysis of Fermentation Conditions

After the center point was identified via the SAT, RSA was used to optimize the first three variables (inoculum size (*X*_1_), fermentation temperature (*X*_2_) and fermentation time (*X*_3_)), screened according to statistically significant influence on *S. fibuligera* Y1402 production determined by PBD (Table 8).

### 2.6. Scaled-Up Fermentation in a 3 L Fermenter

The optimized medium and fermentation conditions were used for an amplification fermentation experiment with *S. fibuligera* Y1402 in a 3 L fermenter (Shanghai Bailun Biological Technology Co., Ltd., Shanghai, China). Due to the rapid growth of this strain under higher oxygen levels, the aeration rate of the fermenter was set to 1.25 vvm based on relevant literature references and experimental conditions [27,28,29]. Considering practical temperature control in the fermenter and cost concerns, the fermentation temperature was set at 25 °C, and the liquid volume was set at 1.8 L. Other conditions were set according to the above optimized conditions.

### 2.7. Biomass Measurement

The biomass of *S. fibuligera* Y1402 was measured using the optical density method (measuring absorbance at 560 nm wavelength) and the plate counting method [30,31].

### 2.8. Data Analysis

Assays were conducted at least in triplicate, and the reported values correspond to the mean value. Statistical differences among the assessed strategies were analyzed using a one-way analysis of variance (ANOVA) (*p* < 0.05) with the Tukey test. SPSS 24.0 (IBM Corp., New York, NY, USA), OriginPro 9.1 (OriginLab, Northampton, MA, USA), Design-expert 11 (Stat-Ease, Inc., Minneapolis, MN, USA), and Excel 2019 (Microsoft, Redmond, NY, USA) were used to process and analyze data.

## 3. Results and Discussion

### 3.1. Optimization of Medium Ingredients for S. fibuligera Y1402

#### 3.1.1. Single-Factor Design of Medium Ingredients

The growth of microorganisms requires nutrients such as water, carbon sources, nitrogen sources, inorganic salts, and growth factors. Different microbes have different requirements for types and concentrations of nutrients due to species differences [20]. This study analyzed the effects of nine basic media on the growth of *S. fibuligera* Y1402. The results (Figure 1a) showed that, apart from basic medium 1, the other media were favorable for the growth of *S. fibuligera* Y1402, especially media 2, 6, 8, and 9. By comparing the composition of different culture media, it could be found that the concentration of carbon sources in medium 1 is low, and the kinds of minerals and growth factors are relatively simple [32]. Therefore, compared with other media, it is not conducive to the growth of yeast Y1402. The iron in media 3, 4, and 5 might prevent the action of yeast Y1402-related enzymes, making them less suitable for yeast growth, even if they all provide a significant amount of carbon and nitrogen sources and an array of nutrients [5,33]. Additionally, due to its low carbon source concentration and very simple minerals and growth nutrients, medium 7 is likely not the ideal basic growth medium for yeast Y1402. Media 2, 6, 8, and 9 contain high concentrations of carbon and nitrogen sources and a variety of other nutrients, which could effectively promote the growth of yeast Y1402. Medium 9 had the most visible impact out of the four media, with a high concentration of readily available carbon sources and an abundance of different nitrogen sources, minerals, and growth stimulants. [34]. Therefore, the researchers considered medium 9 to be the most favorable basic growth medium for *S. fibuligera* Y1402, and conducted subsequent experiments based on this medium.

Carbon sources are the primary energy source for microbial growth and serve as the carbon skeleton for numerous substances in microbial cell metabolism. Sufficient carbon sources are crucial for microbial growth [35]. Different microbial species require different concentrations of carbon sources for their growth and metabolism due to their characteristics. As shown in Figure 1b, the optimal glucose concentration for the growth of *S. fibuligera* Y1402 was 300 g/L. Due to insufficient available energy, *S. fibuligera* Y1402 could not develop in environments with glucose concentrations below 300 g/L. On the other hand, extremely high glucose concentrations hindered the yeast’s metabolism, causing the organism to produce more alcohol and other metabolites or high osmotic pressures, which prevented the organism from developing [36]. This study used a relatively high glucose concentration for two possible reasons. Firstly, *S. fibuligera* Y1402 had higher glucose tolerance due to its long-term survival in the complex environment of Baijiu. Secondly, *S. fibuligera* Y1402 required the longest fermentation time [20,23,24].

Yeast extract contains abundant nitrogen sources such as amino acids and peptides, providing necessary nitrogen sources for yeast cell protoplasm and other materials and various growth factors such as vitamins, trace elements, yeast nucleic acids, yeast polysaccharides and other nutrients. It significantly promotes microbial growth and is the most used and effective component in microbial media [37]. Figure 1c demonstrates that *S. fibuligera* Y1402 achieved the highest cell density at a yeast extract powder concentration of 20 g/L, which aligns with the optimal yeast extract concentration for *Zygosaccharomyces rouxii* as reported by Kang et al. [38]. However, it differed from the optimal yeast extract concentration reported for *H. uvarum* by Zhang et al. [20] and for *Issatchenkia orientalis* by Zheng et al. [23]. This was mainly due to differences in the characteristics of different microorganisms and variations in the basic medium used. *S. fibuligera* Y1402 did not grow well at lower yeast extract concentrations because it did not have enough nitrogen sources and growth factors. On the other hand, high amounts of yeast extract stunted its development by creating an unfavorable carbon-to-nitrogen ratio. Tryptone shares many characteristics with yeast extract due to its high concentration of amino acids, particularly those containing sulphur. It is a popular organic nitrogen source for microorganism fermentation because it includes vitamins and other growth elements needed by microbes [39]. The analysis revealed that the cell density of *S. fibuligera* Y1402 increased as the concentration of tryptone increased. When the tryptone concentration was 50 g/L, the carbon-to-nitrogen ratio in the medium was optimal, resulting in the highest cell density (Figure 1c). However, when the tryptone concentration exceeded 50 g/L, the carbon-to-nitrogen ratio decreased, which was detrimental to the growth of *S. fibuligera* Y1402. Liu et al. [40] reported that the optimal tryptone concentration for culturing *Pichia kudriavzevii* was 3 g/L, while Huang et al. [41] found that the optimal tryptone concentration for culturing zinc-enriched yeast S7 was 26 g/L, which is lower than the optimal concentration for yeast Y1402 in this study. This might be consistent with the reasons for optimizing the yeast extract concentration conditions mentioned above.

Inorganic salts are crucial for developing microorganisms, as they actively contribute to and maintain the integrity of microbial cells. They are parts of certain enzymes and affect how enzymes work in biological processes. They keep osmotic pressure, hydrogen ion concentration, and redox potential stable while microorganisms grow [21,42]. This study examined how different amounts of KH_2_PO_4_, MgSO_4_, and CuSO_4_ affected the growth of *S. fibuligera* Y1402. The ingredients of the optimal basic medium 9 were used to test this. As shown in Figure 1d, the highest cell density was observed when the concentration of KH_2_PO_4_ was 4 g/L, which was similar to the KH_2_PO_4_ concentrations in the optimal growth media for *I. orientalis*, *Kluyveromyces marxianus*, and *P. kudriavzevii* YDF-1 [23,41,43]. Phosphorus is an important component of nucleic acids, phospholipids, and coenzymes. It can participate in the phosphorylation process of carbohydrate metabolism to produce high-energy phosphate compounds such as ATP to store and transport energy. It can also act as a buffer for pH in the growth environment. Therefore, a certain amount of phosphate is beneficial for the growth of yeast Y1402 [23]. Magnesium ions affect sugar conversion rates and coenzyme factor metabolism pathways. They decrease the permeability of the cytoplasmic membrane and enhance the yeast cell’s resistance to metabolites such as ethanol [44,45,46,47,48,49]. They are important elements required for the growth of most microorganisms. As shown in Figure 1e, adding magnesium sulfate benefited the growth of *S. fibuligera* Y1402. However, excessive magnesium concentrations (0.40–6 g/L) had some inhibitory effect on *S. fibuligera* Y1402 growth, although there was no significant difference. This might be because *S. fibuligera* Y1402 did not have high demand for magnesium ions and could tolerate the effects of high concentrations of magnesium sulfate to some extent due to its long-term adaptation to the complex environment of Baijiu fermentation. The optimal concentration of magnesium sulfate in the growth medium for *Saccharomyces boulardii* is 0.80 g/L, while the optimal concentration for *K. marxianus* is 0.30 g/L, and the optimal concentration for *P. kudriavzevii* is 0.50 g/L. These are similar to the results of this study [24,40,43]. Copper, a redox-active metallic element, is indispensable in almost all living cells. It serves as a co-factor in several enzyme structures and metabolic processes, thereby playing a vital part in the development and metabolism of organisms. Microorganisms may use copper to enhance the function of superoxide dis-mutase, a protective enzyme that eliminates excessive free radicals and defends the organism against oxidative damage [50,51]. Figure 1f shows that adding copper sulfate to the medium at levels between 0.01 g/L to 0.50 g/L helped *S. fibuligera* Y1402 grow faster than the group without copper sulfate. However, when the concentration of copper sulfate reached 1 g/L, it inhibited the yeast. This might be attributed to the fact that high concentrations of copper ions decreased the utilization rate of glucose by the yeast, inhibited energy metabolism, damaged cell membrane integrity, and affected the absorption of other metal elements by *S. fibuligera* Y1402 [51,52]. These findings were similar to the results of the study on *P. kudriavzevii* YDF-1 [40].

#### 3.1.2. Plackett–Burman Design of Medium Ingredients

The experimental results are shown in Table 3. Based on the analysis of variance and significance (Appendix A), the results showed that different nutritional components had different effects on the growth of yeast Y1402. Among them, the concentration of yeast extract had a negative effect on the growth of yeast Y1402, while the concentrations of other nutritional components showed a positive effect. Differential analysis showed that of the six nutritional substances, glucose, yeast extract, and KH_2_PO_4_ had the most significant impacts on the growth of *S. fibuligera* Y1402. The other three nutritional substances did not have any significant impacts. Subsequently, the concentrations of glucose, yeast extract, and KH_2_PO_4_ were further optimized, while the concentrations of peptone, MgSO_4_, and CuSO_4_ were selected based on the optimized levels from the SFD results.

#### 3.1.3. Steepest Ascent Test of Medium Ingredients

The design and results of the SAT are shown in Table 4. The results showed that as the experimental groups designed according to the SAT increased, the cell density of yeast Y1402 exhibited a parabolic trend. In the fourth experimental group, the *S. fibuligera* Y1402 cell density was the highest, reaching 46.68 OD_560nm_. Therefore, the concentrations of each medium ingredient corresponding to the fourth experimental group were selected as the central points for the factors involved in the subsequent RSA.

#### 3.1.4. Response Surface Analysis of Medium Ingredients

Based on the results of the PBD and SAT, a three-factor and three-level RSA was designed using the Box–Behnken design with glucose (*X*_1_), yeast extract (*X*_2_), and KH_2_PO_4_ (*X*_3_) as factors (Table 5). The experimental results were analyzed using Design-Expert software, and a multivariate quadratic regression equation between *S. fibuligera* Y1402 cell density and glucose (*X*_1_), yeast extract (*X*_2_), and KH_2_PO_4_ (*X*_3_) was obtained as follows:Y = −8.16 + 0.03935*X*_1_ + 0.1458*X*_2_ + 0.395*X*_3_ − 0.00006*X*_1_^2^ − 0.002977*X*_2_^2^ −0.0534*X*_3_^2^ − 0.000048*X*_1_*X*_2_ + 0.000840*X*_1_*X*_3_ − 0.00755*X*_2_*X*_3_

According to the analysis of variance (Appendix A), the *p*-value of the multivariate quadratic regression equation was less than 0.001, indicating that the regression equation was highly significant. Additionally, the *R*^2^ value of the regression equation was 96.48%, indicating a good fit of the equation to the experiment with low error. The lack-of-fit term *p*-value was 0.084 > 0.05, indicating that there was a good correlation between the actual values and predicted values of the equation.

Variance analysis revealed that various factors influence yeast Y1402 growth in the following order: KH_2_PO_4_ concentration, glucose concentration, and yeast extract concentration. Primary terms A and C are extremely significant (*p* < 0.01), while primary term B is not significant. Additionally, A^2^, B^2^, and C^2^ have extremely significant curve effects on the growth of yeast Y1402. There are also significant interactions between glucose concentration and KH_2_PO_4_ concentration, as well as between yeast extract concentration and KH_2_PO_4_ concentration. However, the interaction between glucose concentration and yeast extract concentration is not significant. We used the regression equation to generate response surface plots (Figure 2) to demonstrate the interplay between the various factors visually. As shown in Figure 2a, cell density first increases and then decreases as glucose or yeast extract concentration increases. The optimal growth conditions for yeast Y1402 occur when the glucose concentration is 360.61 g/L and the yeast extract concentration is 14.65 g/L. Similarly, Figure 2b demonstrates that cell density also exhibits a trend of increasing and then decreasing as glucose concentration or KH_2_PO_4_ concentration increases. The optimal growth conditions for yeast Y1402 are achieved when the glucose concentration is 360.61 g/L and KH_2_PO_4_ concentration is 5.49 g/L. Finally, Figure 2c indicates that cell density also follows a trend of increasing and then decreasing as yeast extract concentration or KH_2_PO_4_ concentration increases. The highest cell density is achieved when the yeast extract concentration is 14.65 g/L and KH_2_PO_4_ concentration is 5.49 g/L. By solving this equation using Design-Expert software, the optimal culture medium for the growth of yeast Y1402 was determined to have glucose, yeast extract, and KH_2_PO_4_ concentrations of 360.61 g/L, 14.65 g/L, and 5.49 g/L, respectively. In this optimized medium, the cell density of *S. fibuligera* Y1402, measured as OD_560nm_, was found to be 53.95, which was very close to the predicted value of 53.75. The cell density in the optimized medium was improved by 20% compared to the highest level achieved with SFD (OD_560nm_ = 44.90).

### 3.2. Optimization of Fermentation Conditions for S. fibuligera Y1402

#### 3.2.1. Single-Factor Design of Fermentation Conditions

Changes in the pH value of the microbial growth environment can affect the existence of nutrients required by microorganisms, leading to differences in absorption and utilization efficiency of nutrients by microbial cells. Therefore, conditions are more conducive to the growth of microorganisms under a suitable pH value, especially an initial pH value [53]. It has been found that the optimal initial pH value for *S. fibuligera* Y1402 growth is in the range of 6.0–6.5, and excessively low or high initial pH values could both have inhibitory effects on yeast Y1402 (Figure 3a). Studies have shown that the optimal initial pH for the growth of most yeast species ranges from 4.50 to 6.50, depending mainly on the yeast strain and its long-term survival environment [23,24,43].

The inoculum size is one of the most important metrics for controlling microorganisms utilized in industrial production. It impacts manufacturing cost savings, the fermentation cycle’s duration, the fermentation objective’s output, and the danger of microbial contamination. A low inoculum size leads to a longer lag phase, an extended fermentation cycle and a lower yield, with an increased risk of contamination. On the other hand, a large inoculum results in increased production costs and abnormal microbial metabolism due to the rapid consumption of nutrients and oxygen. Thus, an appropriate inoculum size is necessary to ensure efficient microbial fermentation [54]. When optimizing the inoculum size for *S. fibuligera* Y1402, it was found that a 1% inoculum size resulted in the highest biomass for this yeast (Figure 3b). This is close to the inoculation size of *I. orientalis* and lower than the optimal inoculation size of *S. boulardii* [23,24].

Yeast is a facultative anaerobic microorganism. Under aerobic conditions, it can quickly break down sugar into carbon dioxide and water, which promotes yeast growth [55]. Two critical variables in flask culture for controlling the amount of dissolved oxygen during fermentation are liquid volume and rotation speed. Thus, the dissolved oxygen level during fermentation and the development and fermentation of microorganisms may be efficiently controlled by optimizing the liquid volume and rotation speed. [56]. Figure 3c demonstrates that the cell density of *S. fibuligera* Y1402 increased as the liquid volume decreased, suggesting that higher oxygen content promoted yeast growth. Consequently, a liquid volume of 12.5 mL/250 mL was chosen for subsequent experiments. Furthermore, Figure 3c also reveals that a rotation speed of 90 rpm led to a low level of dissolved oxygen during fermentation, which had a negative impact on the growth of *S. fibuligera* Y1402. When the rotation speed was increased to 120 rpm, the dissolved oxygen level in the medium was sufficient to meet the growth requirements of yeast. However, further increasing the rotation speed did not significantly increase yeast growth. This might be because higher rotation speeds not only increased the dissolved oxygen content in the medium but also increased the shear force, which had a suppressive effect on the growth of *S. fibuligera* Y1402. Therefore, when both dissolved oxygen and mechanical shear force increase, the cell density of yeast Y1402 does not change significantly. Considering the cost issue, a rotation speed of 120 rpm is preferred in the later stage, and is consistent with the optimal rotation speed required for the growth of *I. orientalis* [23].

Temperature can affect the fluidity of microbial cell membranes, which is an important parameter that affects microbial growth. Each microorganism has its own optimal growth temperature [57]. As shown in Figure 3a, the cell density of *S. fibuligera* Y1402 exhibited an increasing and then decreasing trend with increasing fermentation temperature. Among these temperatures, the highest cell density was achieved at 20 °C with an OD_560nm_ value of 58.2, which was lower than the optimal growth temperature (25 °C) reported for this strain in YPD medium and the optimal growth temperature reported for other *S. fibuligera* strains [58]. This could be due to the osmotic pressure from high glucose concentrations reducing the impact on the yeast cells at lower temperatures.

In batch fermentation, studying the microbial growth curve is crucial. When fermentation is shorter, microbial growth may be in a lag or logarithmic growth phase, resulting in low cell density. Conversely, when fermentation is longer, microbial growth stagnates, and cell death occurs due to nutrient depletion, leading to a decrease in cell density caused by autolysis. As shown in Figure 3d, the highest biomass of *S. fibuligera* Y1402 was observed at two days of fermentation, indicating that the yeast was in a stable growth phase around this time. The cultivation time was longer than that of *H. uvarum* and *K. marxianus*. The features of the yeast strains, the nutritional components of the culture medium, and the culture conditions all contribute to this phenomenon [20,43].

#### 3.2.2. Plackett–Burman Design of Fermentation Conditions

The experimental results of PBD are shown in Table 6. The results of the analysis of variance (Appendix A) showed that the inoculum size, fermentation time, and fermentation temperature all had significant impacts on the growth of *S. fibuligera* Y1402, which is why these strains were chosen for further studies. However, the rotation speed and initial pH had no significant effect on its growth, and were set to the optimal levels obtained from the SFD.

#### 3.2.3. Steepest Ascent Test of Fermentation Conditions

According to the results of the Plackett–Burman test, the three factors (inoculum size, culture time, and culture temperature) that significantly affect the growth of yeast Y1402 have positive correlation coefficients with the response value, indicating that they are positively correlated with the growth of yeast Y1402. Therefore, the levels of these factors should be increased when designing the SAT. The SAT scheme was designed by combining the SFD results with practical operation experience. The experimental results obtained based on this design are shown in Table 7. The maximum cellular concentration of *S. fibuligera* Y1402 was attained during the third set of trials, resulting in an optical density (OD) value of 86.40 at a wavelength of 560 nm. Thus, the factors and their levels corresponding to the third batch of experiments were selected for the subsequent RSA.

#### 3.2.4. Response Surface Analysis of Fermentation Conditions

Based on the results of the SAT, a Box–Behnken experiment with three factors (inoculum size (*X*_1_), fermentation temperature (*X*_2_), and fermentation time (*X*_3_)) at three levels was used for the RSA, and the results are shown in Table 8. Design-Expert software was used to process the experimental results, and the quadratic regression equation between the cell density of *S. fibuligera* Y1402 and the three factors was as follows:Y = 470.1 + 98.3*X*_1_ + 39.92*X*_2_ + 76.16*X*_3_ − 22.33*X*_1_^2^ − 0.8645*X*_2_^2^ − 13.36*X*_3_^2^ −1.762*X*_1_*X*_2_ − 5.70*X*_1_*X*_3_ − 0.575*X*_2_*X*_3_

According to the analysis of variance (Appendix A), the *p*-value was <0.001, and the lack-of-fit term *p* = 0.087 > 0.05, indicating that this regression equation was highly significant. Additionally, the coefficient of determination (*R*^2^) of the quadratic regression equation was 98.61%, indicating a good fit between the equation and the experimental results. In the regression equation, the *p*-values of the first-order terms are all less than 0.01, indicating an extremely significant level. The RSA chart and contour plot of the three factors were drawn based on the above regression equation, shown in Figure 4. According to the contour plot (Figure 4), the cell density of yeast first increased as the quantities of the three variables increased and then declined after reaching an ideal value. This suggests that an ideal mix of these elements maximizes cell density.

The optimal combination of factors was determined to be an inoculum size of 1.10%, fermentation time of 53.50 h, and fermentation temperature of 21 °C. Under these conditions, the predicted cell density of *S. fibuligera* Y1402 was an OD_560nm_ value of 90.02. To validate the model, a verification experiment was conducted using these optimal conditions, and the measured cell density was found to be 90.11 OD_560nm_, which was close to the predicted value. This further confirmed the reliability of the model.

### 3.3. Scaled-Up Fermentation in a 3 L Fermenter

Using the optimized medium ingredients and fermentation conditions, scaled-up fermentation of *S. fibuligera* Y1402 was conducted in a 3 L fermenter. A fermentation cycle of 72 h was used, with samples taken every 8 h to measure the cell density, pH, and residual sugar concentration in the medium. The growth curve of *S. fibuligera* Y1402 and the change in residual sugar concentration can be seen in Figure 5.

From the changes in residual sugar concentration, it can be observed that after 8 h of fermentation, the glucose concentration sharply decreased and reached a very low level at 32 h. During this period, *S. fibuligera* Y1402 grew rapidly with increasing fermentation time and reached a proliferation phase at 32 h. This phase demanded a large amount of carbon, but the carbon source in the medium was unable to meet the growth needs of yeast, resulting in growth inhibition. However, after a short period of stagnation, the yeast resumed growth, and the overall growth pattern exhibited a two-stage growth curve. The first stage involved glucose fermentation for growth. After the glucose was completely consumed, the yeast utilized the ethanol produced during fermentation as a carbon source for the second growth stage. Simply increasing the glucose concentration would lead to the glucose effect, resulting in ethanol accumulation and inhibition of cell growth. To overcome this problem, glucose can be added continuously to avoid the accumulation of glucose to toxic levels and prevent the cells from experiencing starvation. In addition, providing a favorable oxygen environment for aerobic metabolism can lead to a faster specific growth rate, resulting in higher yeast production.

## 4. Conclusions

Through the analysis of SFD, PBD, SAT, and RSA tests, the optimal medium formulation and fermentation conditions for the high-density fermentation of *S. fibuligera* Y1402 were determined. Fermentation under the optimal conditions involved an initial pH of 6, an inoculum size of 1.10%, a rotation speed of 120 r/min, a fermentation temperature of 21 °C, a liquid volume of 12.50 mL/250 mL, and a fermentation time of 53.50 h, with the optimal medium including a glucose concentration of 360.61 g/L, a peptone concentration of 50 g/L, a yeast extract concentration of 14.65 g/L, KH_2_PO_4_ concentration of 5.49 g/L, MgSO_4_ concentration of 0.40 g/L, and CuSO_4_ concentration of 0.01 g/L. The OD_560nm_ value reached 90.11 and the total colony count reached 5.50 × 10^9^ CFU/mL. This study has guiding significance for the future industrial production and application of *S. fibuligera*.

## Figures and Tables

**Figure 1 foods-13-01546-f001:**
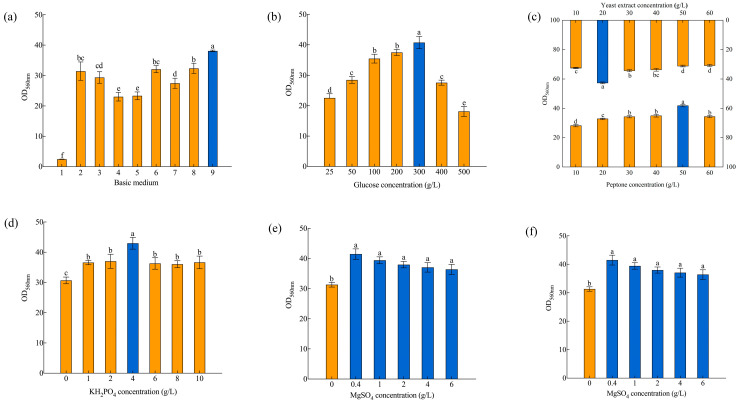
Optimization of medium ingredients via SFD. (**a**) Type of basic culture medium, (**b**) glucose concentration, (**c**) peptone concentration and yeast extract concentration, (**d**) KH_2_PO_4_ concentration, (**e**) MgSO_4_ concentration, (**f**) CuSO_4_ concentration. The same letters in a column indicate that the data do not differ significantly at 5% probability using Tukey’s test. The blue bar means better conditions for the growth of *S. fibuligera* Y1402.

**Figure 2 foods-13-01546-f002:**
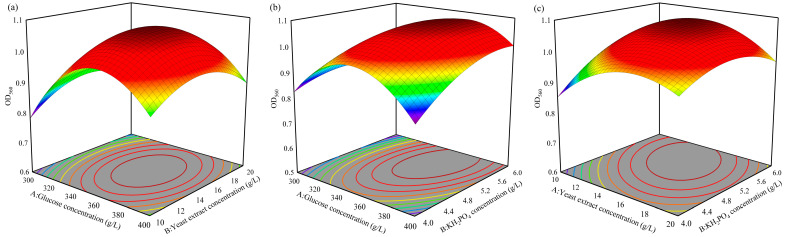
Response surface and contour map of the interaction of various factors affecting the cell density of *S. fibuligera* Y1402. (**a**) Interaction between glucose concentration and yeast extract concentration; (**b**) interaction between glucose concentration and KH_2_PO_4_ concentration; (**c**) interaction between yeast extract concentration and KH_2_PO_4_ concentration. The color change from blue to red corresponds to an increase in responses.

**Figure 3 foods-13-01546-f003:**
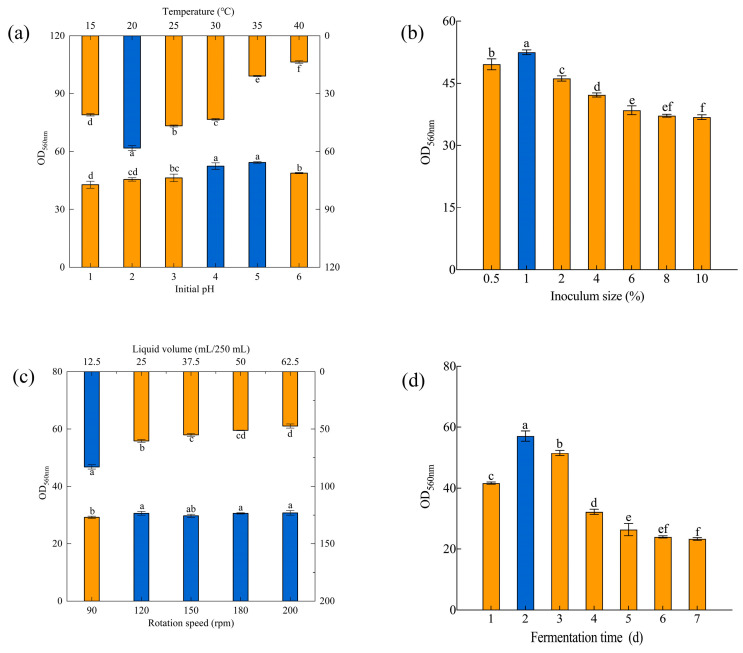
Optimization of fermentation conditions by SFD: (**a**) Initial pH and fermentation temperature, (**b**) inoculum size, (**c**) rotation speed and liquid volume, (**d**) fermentation time. The same letters in a column indicate that the data do not differ significantly at 5% probability using Tukey’s test. The blue bar means better conditions for the growth of *S. fibuligera* Y1402.

**Figure 4 foods-13-01546-f004:**
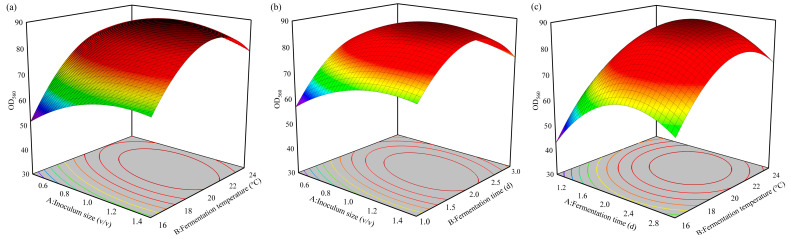
Response surface and contour map of the interactions of various factors affecting the cell density of *S. fibuligera* Y1402. (**a**) Interaction between inoculum size and fermentation temperature, (**b**) interaction between inoculum size and fermentation time, (**c**) interaction between fermentation time and fermentation temperature. The color change from blue to red corresponds to an increase in responses.

**Figure 5 foods-13-01546-f005:**
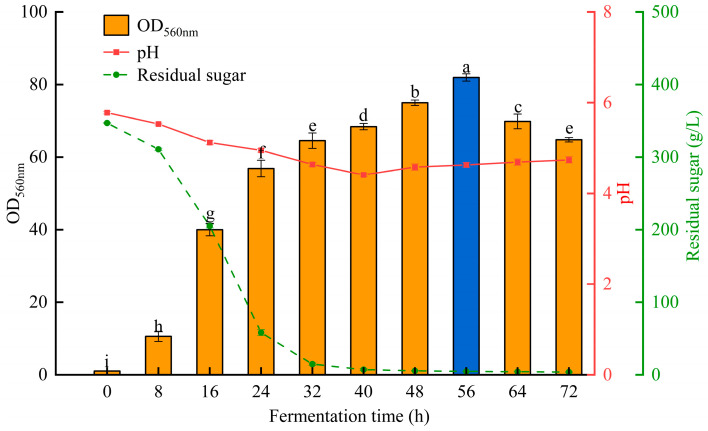
The growth curve of *S. fibuligera* Y1402 in a 3 L fermenter. The same letters in a column indicate that the data do not differ significantly at 5% probability using Tukey’s test. The blue bar means the best condition for the growth of *S. fibuligera* Y1402.

**Table 1 foods-13-01546-t001:** Ingredients of culture media (g/L).

Ingredient	Basic Medium								
	1	2	3	4	5	6	7	8	9
Glucose	50	-	50	70	50	40	40	-	100
Sucrose	-	76	-	20	-	-	-	67	-
Maltose	-	-	-	20	-	-	-	-	-
Peptone	-	-	-	-	-	45	40	-	10
Tryptone	-	36	20	16	30	-	-	-	-
Yeast extract	-	-	10	15	-	-	-	30	10
Urea	5	-	-	-	-	-	-	-	-
Glycine	0.5	-	-	-	-	-	-	-	-
(NH_4_)_2_SO_4_	0.5	1	5	-	-	-	-	-	-
KH_2_PO_4_	-	2	1	1	1.2	0.6	3.9	4.13	2
K_2_HPO_4_	-	-	-	1	-	-	-	-	-
MgSO_4_	-	1.5	-	5	0.02	0.8		0.3	0.4
KCl	-	-	0.4	-	-	-	-	-	-
CaCl_2_	-	-	1.1	-	-	-	-	-	-
FeCl_3_	-	-	0.0025	-	-	-	-	-	-
ZnSO_4_	-	-	-	0.01	0.002	-	-	-	-
FeSO_4_	-	-	-	0.006	0.002	-	-	-	-
CuSO_4_	-	-	-	0.01	0.01	-	-	-	0.001
KNO_3_	-	-	-	-	-	14	-	-	-
Corn plasm	-	-	-	-	-	15	-	16	-
Yeast nutrient	-	-	-	-	-	3	-	-	-

**Table 2 foods-13-01546-t002:** Factors and levels for SFD.

Factor	Level
Glucose concentration (g/L)	100, 200, 300, 400 and 500
Yeast extract concentration (g/L)	10, 20, 30, 40, 50 and 60
Peptone concentration (g/L)	10, 20, 30, 40, 50 and 60
KH_2_PO_4_ concentration (g/L)	1, 2, 4, 8 and 10
MgSO_4_ concentration (g/L)	0.4, 1, 2, 4 and 6
CuSO_4_ concentration (g/L)	0.01, 0.05, 0.1, 0.5 and 1
Initial pH	4.5, 5.0, 5.5, 6.0, 6.5 and 7.0
Inoculum size (*v*/*v*)	0.5%, 1%, 2%, 4%, 6%, 8% and 10%
Liquid volume (mL/250 mL)	12.5, 25, 37.5, 50 and 62.5
Rotation speed (r/min)	90, 120, 150, 180 and 200
Fermentation temperature (°C)	15, 20, 25, 30, 35 and 40
Fermentation time (d)	1, 2, 3, 4, 5, 6 and 7

**Table 3 foods-13-01546-t003:** PBD and results for medium ingredients.

Test Number	*X* _1_	*X* _2_	*X* _3_	*X* _4_	*X* _5_	*X* _6_	OD_560nm_
1	1	−1	1	−1	−1	−1	40
2	1	1	−1	1	1	−1	36.5
3	−1	−1	−1	−1	−1	−1	41.95
4	1	−1	1	1	−1	1	25
5	−1	−1	1	1	1	−1	30
6	0	0	0	0	0	0	43.5
7	0	0	0	0	0	0	44
8	1	1	−1	1	−1	−1	40.83
9	−1	1	−1	−1	−1	1	25
10	1	1	1	−1	1	1	41
11	−1	1	1	−1	1	−1	35
12	0	0	0	0	0	0	41.5
13	−1	1	1	1	−1	1	35
14	1	−1	−1	−1	1	1	41.63
15	−1	−1	−1	1	1	1	36

**Table 4 foods-13-01546-t004:** Experimental designs and results of SAT in OD_560nm_ for medium ingredients.

Test Number	Variable			OD_560nm_
	Glucose Concentration (g/L)	Yeast Extract Concentration (g/L)	KH_2_PO_4_ Concentration (g/L)	
1	200	30	2	33.37
2	250	25	3	38.12
3	300	20	4	40.10
4	350	15	5	46.68
5	400	10	6	39.90

**Table 5 foods-13-01546-t005:** RSA and responses of the dependent variables for medium ingredients.

Test Number	Glucose Concentration (g/L)		Yeast Extract Concentration (g/L)		KH_2_PO_4_ Concentration (g/L)		OD_560nm_
	*X* _1_	Code *X*_1_	*X* _2_	Code *X*_2_	*X* _3_	Code *X*_3_	Y
1	300	−1	15	0	4	−1	39.61
2	400	1	15	0	4	−1	40.53
3	350	0	20	1	4	−1	48.52
4	350	0	20	1	6	1	47.45
5	300	−1	20	1	5	0	40.75
6	350	0	15	0	5	0	53.24
7	400	1	10	−1	5	0	44.15
8	350	0	10	−1	4	−1	42.17
9	400	1	20	1	5	0	42.25
10	300	−1	10	−1	5	0	40.26
11	350	0	15	0	5	0	51.45
12	350	0	15	0	5	0	53.25
13	350	0	10	−1	6	1	48.60
14	350	0	15	0	5	0	53.14
15	350	0	15	0	5	0	53.40
16	300	−1	15	0	6	1	41.09
17	400	1	15	0	6	1	50.41

**Table 6 foods-13-01546-t006:** PBD and results for fermentation conditions.

Test Number	*X* _1_	*X* _2_	*X* _3_	*X* _4_	*X* _5_	OD_560nm_
1	5	1.5	150	15	3	65.28
2	7	0.5	90	25	3	65.50
3	7	1.5	90	25	3	71.77
4	6	1	120	20	2	89.30
5	7	0.5	150	15	1	2.37
6	5	0.5	150	25	3	60.17
7	7	1.5	90	15	1	6.46
8	6	1	120	20	2	87.63
9	7	1.5	150	15	3	89.96
10	5	1.5	90	25	1	56.33
11	5	0.5	90	15	1	3.38
12	7	0.5	150	25	1	75.20
13	6	1	120	20	2	87.73
14	5	0.5	90	15	3	63.26
15	5	1.5	150	25	1	70.24

**Table 7 foods-13-01546-t007:** Experimental designs and results of SAT in OD_560nm_ for fermentation conditions.

Test Number	Variable			OD_560nm_
	Inoculum Size (*v*/*v*)	Fermentation Temperature (°C)	Fermentation Time (h)	
1	0.50	15.0	24	23.78
2	0.75	17.5	36	37.82
3	1.00	20.0	48	86.40
4	1.25	22.5	60	83.77
5	1.50	25.0	72	66.10

**Table 8 foods-13-01546-t008:** RSA and responses of the dependent variables for fermentation conditions.

Test Number	Inoculum Size (*v*/*v*)		Fermentation Temperature (°C)		Fermentation Time (h)		OD_560nm_
	*X* _1_	Code *X*_1_	*X* _2_	Code *X*_2_	*X* _3_	Code *X*_3_	Y
1	1.0	0	20	0	2	0	87.3
2	1.0	0	20	0	2	0	87.8
3	0.5	−1	20	0	3	1	74.6
4	1.0	0	20	0	2	0	87.7
5	1.5	1	16	−1	2	0	67.3
6	0.5	−1	24	1	2	0	76.4
7	1.0	0	20	0	2	0	88.0
8	0.5	−1	16	−1	2	0	53.8
9	1.0	0	20	0	2	0	87.9
10	1.5	1	24	1	2	0	75.8
11	1.5	1	20	0	1	−1	68.7
12	1.0	0	16	−1	1	−1	42.3
13	1.0	0	16	−1	3	1	55.4
14	1.0	0	24	1	1	−1	70.2
15	1.5	1	20	0	3	1	76.5
16	1.0	0	24	1	3	1	74.2
17	0.5	−1	20	0	1	−1	55.4

## Data Availability

The raw data supporting the conclusions of this article will be made available by the authors on request. The data are not publicly available due to privacy restrictions.

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
