# Peer review of "Optimization of High-Density Fermentation Conditions for Saccharomycopsis fibuligera Y1402 through Response Surface Analysis"

_foods, 2024, doi:10.3390/foods13101546_

Round 1

Reviewer 1 Report

Comments and Suggestions for Authors

The article titled needs some adjustments ” Optimization of High-density Fermentation Conditions for 2 Saccharomycopsis fibuligera Y1402”

Abstract

In order to provide a more comprehensive understanding of the subject matter, it is recommended that the following information be included in the abstract:

(i) A brief context of the biotechnological significance of the microbial strain, which justifies the rationale behind optimizing the fermentation process.

(ii) A conclusion, which should include a percentage comparison of the results obtained before and after the fermentation process.

Materials and methods

2.1. Strains and Materials

2.2. Medium preparation

L(88). Adjust "natural pH" to a correct pH value.

-          It is recommended that, when utilizing multiple culture media, they be presented in a tabular format to enhance the clarity and comprehension of the reader.

2.3. Yeast activation and basic medium selection

L(110). What was the final volume of the new medium to which the reactivated strain was transferred?

L(112). As the name of the strain has already been mentioned above, it is not necessary to include the full name of the strain in order to avoid redundancies in the text.

L(113-114). Please provide an explanation as to why the inoculum volume was changed from 0.5% to 1%. Additionally, confirm whether the use of the 560 nm wavelength is appropriate for the specific strain in question. If this data is derived from an experimental value, please provide a brief description of the methodology employed.

References are required.

2.4. Optimization of medium ingredientes

2.4.1. Single-factor design (SFD)

2.4.2. Plackett-Burman design (PBD)

L(127). What were the numerical values of the -1 and +1 levels?

2.5.2. Plackett-Burman design (PBD)

L(159). What were the numerical values of the -1 and +1 levels?

3. Results and Discussion

3.1.1. Single-factor design (SFD) of medium ingredients

L(302). It is recommended that the size of this series of graphs be increased in order to facilitate a more comprehensive appreciation of the data presented. Furthermore, it is advised that a brief description of each graph be included (a, b, c, etc.).

3.1.4. Response Surface Analysis (RSA) of medium ingredientes

L(365). It is recommended that the size of this series of graphs be increased in order to facilitate a more comprehensive appreciation of the data presented. Furthermore, it is advised that each graph be accompanied by a brief description that provides a succinct overview of the behavior exhibited by the response (a, b, c, etc.).

3.2.1. Single-factor design (SFD) of fermentation conditions

L(431). It is recommended that the size of this series of graphs be increased in order to facilitate a more comprehensive appreciation of the data presented. Furthermore, it is advised that a brief description of each graph be included (a, b, c, d.).   

3.2.4. RSA of fermentation conditions

L(477). It is recommended that the size of this series of graphs be increased in order to facilitate a more comprehensive appreciation of the data presented. Furthermore, it is advised that each graph be accompanied by a brief description that provides a succinct overview of the behavior exhibited by the response (a, b, c, etc.).

Please strain names in italics

Closing remarks

In the data presented as O.D. (optical density),

-          it is recommended that these data be replaced with quantifiable values, such as CFU/mL or biomass g/L, in order to facilitate more accurate analysis of yield and productivity. Optical density provides an indication of the number of cells in a culture but lacks a unit of measurement.

What was the point of planning so many experiments? The whole thing could have been reduced to a Plackett–Burman planning, and an optimization with a box behnken design

Author Response

20240424

Re: Manuscript ID. Foods-2977575 “Optimization of High-density Fermentation Conditions for Saccharomycopsis fibuligera Y1402”

Thank you very much for providing us with helpful suggestions again. In the following, we have made modifications and replies point-by-point according to your suggestions. Revisions in the manuscript are highlighted in blue.

We look forward to hearing from you at your early convenience.

Yours sincerely,

Guangsen Fan

Address: School of Food and Health, Beijing Technology and Business University, Beijing 100048, China

Tel: +86 13811497684

Responses to the comments

  1. The article titled needs some adjustments “Optimization of High-density Fermentation Conditions for Saccharomycopsis fibuligera Y1402”

Response: Thank you very much for your comment. We have adjusted the title to “Optimization of High-density Fermentation Conditions for Saccharomycopsis fibuligera Y1402 through Response Surface Methodology” (Page 1, line 2-4).

  1. Abstract

In order to provide a more comprehensive understanding of the subject matter, it is recommended that the following information be included in the abstract:

(i) A brief context of the biotechnological significance of the microbial strain, which justifies the rationale behind optimizing the fermentation process.

(ii) A conclusion, which should include a percentage comparison of the results obtained before and after the fermentation process.

Response: Thank you very much for your comment. According to the comments, the description of the biotechnological significance of the microbial strain and the comparison of the results was added in the abstract.

Page 1, line 17-22:

Through extensive research on various traditional fermented foods, scientists have discovered the significant contribution of Saccharomycopsis fibuligera. This microorganism is responsible for producing a range of enzymes, including amylase and protease, as well as important flavor com-pounds such as β-phenylethanol and phenyl acetate, in fermented foods. Nevertheless, the absence of a high-density fermentation culture for this strain has somewhat impeded its industrial application process.  

Page 1, line 17-22:

When fermentation was conducted under the optimized medium and conditions, the total colony count achieved a remarkable value of 5.50×10⁹ CFU/mL, exhibiting a substantial increase of nearly 31 times compared to the value in the optimal culture medium. This significant advancement offers valuable insights and a reference for the industrial-scale production of S. fibuligera.

  1. Materials and methods

3.1 L(88). Adjust "natural pH" to a correct pH value.

Response: Thank you very much for your comment. Under your advice, we have reconfigured the YPD medium, measured its pH value, and made adjustments based on the measurement.

Page 2, line 93-94:

Yeast extract peptone dextrose (YPD) medium: yeast extract powder 10 g/L, pep-tone 20 g/L, and glucose 20 g/L, pH 5.8;

3.2 It is recommended that, when utilizing multiple culture media, they be presented in a tabular format to enhance the clarity and comprehension of the reader.

Response: Thank you very much for your comment. According to the comments, the multiple medium ingredients was presented in a tabular format (Table 1).

Page 3, line 97-98:

Table 1. Ingredients of Culture Medium (g/L)

Ingredient

Basic Medium

1

2

3

4

5

6

7

8

9

Glucose

50

-

50

70

50

40

40

-

100

Sucrose

-

76

-

20

-

-

-

67

-

maltose

-

-

-

20

-

-

-

-

-

Peptone

-

-

-

-

-

45

40

-

10

Tryptone

-

36

20

16

30

-

-

-

-

Yeast extract

-

-

10

15

-

-

-

30

10

Urea

5

-

-

-

-

-

-

-

-

Glycine

0.5

-

-

-

-

-

-

-

-

(NH4)2SO4

0.5

1

5

-

-

-

-

-

-

KH2PO4

-

2

1

1

1.2

0.6

3.9

4.13

2

K2HPO4

-

-

-

1

-

-

-

-

-

MgSO4

-

1.5

-

5

0.02

0.8

0.3

0.4

KCl

-

-

0.4

-

-

-

-

-

-

CaCl2

-

-

1.1

-

-

-

-

-

-

FeCl3

-

-

0.0025

-

-

-

-

-

-

ZnSO4

-

-

-

0.01

0.002

-

-

-

-

FeSO4

-

-

-

0.006

0.002

-

-

-

-

CuSO4

-

-

-

0.01

0.01

-

-

-

0.001

KNO3

-

-

-

-

-

14

-

-

-

Corn plasm

-

-

-

-

-

15

-

16

-

Yeast nutrient

-

-

-

-

-

3

-

-

-

3.3 Yeast activation and basic medium selection

L(110). What was the final volume of the new medium to which the reactivated strain was transferred?

Response: Thank you very much for your comment. The final volume of the new medium was 50 mL.

Page 3, line 102-103:

The activated yeast was then transferred to 50 mL of YPD medium again, with an in-oculum size of 0.5 %.

3.4 L(112). As the name of the strain has already been mentioned above, it is not necessary to include the full name of the strain in order to avoid redundancies in the text.

Response: Thank you very much for your comment. According to the comments, we changed its name to an abbreviation when the strain appeared again.

3.5 L(113-114). Please provide an explanation as to why the inoculum volume was changed from 0.5% to 1%. Additionally, confirm whether the use of the 560 nm wavelength is appropriate for the specific strain in question. If this data is derived from an experimental value, please provide a brief description of the methodology employed. References are required.

Response: Thank you very much for your comment. The inoculation amount of 1% here is set as the initial inoculation amount for exploring high-density fermentation culture, while the inoculation amount of 0.5% is to activate the strain for the second time. Although there is a certain relationship between the two, we only set a basic inoculation amount for exploring high-density culture. We need to optimize the inoculation amount factor in high-density culture in the next. That is to say, we didn't pay attention to the connection between the two at the early stage of the experiment. One is for strain activation, and the other is just an initial condition set for optimizing culture conditions.

We conducted the measurement of yeast density at the wavelength of 560 nm based on relevant research reports. In fact, there are many wavelengths for the measurement of yeast density, such as 488 nm, 540 nm, 560 nm, 570 nm, 600 nm, 620 nm, 660 nm and 750 nm, and 560 nm and 600 nm are commonly used. Referring to relevant research and the team's usual methods, we selected 560 nm for yeast and 600 nm for bacteria to measure turbidity to reflect strain concentration. Although it is possible that the selected wavelength may not accurately reflect the number of strain cells, and this method itself is not very accurate for measuring the number of strain cells, it can quickly reflect the effect of culture conditions on strain growth. In addition, from our previous research on this strain, we found that there was a good positive correlation between the OD560 number of the strain and the number of viable cells measured by colony counting. Therefore, we selected the turbidity measured at this wavelength to reflect the growth of yeast, which can be supported by data on the effect of optimized culture time on strain cells growth. In order to more scientifically demonstrate the scientific nature of our research, we have supplemented the description of the method and relevant references as per your advice.

References:

Van Hamersveld, E.H., Van der Lans, R.G.J.M.; Luyben, K.C.A.M. Quantification of brewers' yeast flocculation in a stirred tank: Effect of physical parameters on flocculation. J. Biotechnol Bioeng. 1997, 56(2): 190-200. DOI: 10.1002/(sici)1097-0290(19971020)56:2<190::aid-bit8>3.0.co;2-k.

Gregory, M.; Thornhill, N. The effects of aeration and agitation on the measurement of yeast biomass using a laser turbidity probe. J. Bioproc Biosyst Eng. 1997, 16: 339-344. DOI: 10.1007/s004490050332.

Galgiani, J.N.; Stevens, D.A. Antimicrobial Susceptibility Testing of Yeasts: a Turbidimetric Technique Independent of Inoculum Size. J. Antimicrob Agents Ch. 1976, 10(4): 721-726. DOI: 10.1128/aac.10.4.721.

Yamashita, S.; Matsunaga, K.; Miura, H.; Morikawa, M.; Shimizu, K.; Matsubara, M.; Kobayashi, T. Use of a novel turbidimeter to monitor microbial growth and control glucose concentration. J. J Chem Technol Biot. 1987, 40(3): 203-213. DOI: 10.1002/jctb.280400306.

Smit, G.; Straver, M.H.; Lugtenberg, B.J.; Kijne, J.W. Flocculence of Saccharomyces cerevisiae cells is induced by nutrient limitation, with cell surface hydrophobicity as a major determinant. J. Appl Environ Microbiol. 1992, 58(11): 3709-3714. DOI: 10.1128/aem.58.11.3709-3714.1992.

Mertens, L.; Van Derlinden, E.; Dang, T. D. T.; Cappuyns, A.M.; Vermeulen, A.; Debevere, J.; Van Impe, J.F. On the critical evaluation of growth/no growth assessment of Zygosaccharomyces bailii with optical density measurements: liquid versus structured media. J. Food microbiol. 2011, 28(4): 736-745. DOI: 10.1016/j.fm.2010.05.032.

Schwinn, M.; Durner, D.; Delgado, A.; Fischer, U. Distribution of yeast cells, temperature, and fermentation by-products in white wine fermentations. J. Am J Enol Viticult. 2019, 70(4): 339-350. DOI: 10.5344/ajev.2019.18092.

Spiden, E.M.; Scales, P.J.; Kentish, S.E.; Martin, G.J.O. Critical analysis of quantitative indicators of cell disruption applied to Saccharomyces cerevisiae processed with an industrial high pressure homogenizer. J. Biochem Eng J. 2013, 70: 120-126. DOI: 10.1016/j.bej.2012.10.008.

Castro, G.R; Andribet E.P.; Ducrey, L.M.; Garro, O.A.; Siñeriz, F. Modelling and operation of a turbidity-meter for on-line monitoring of microbial growth in fermenters. J. Process biochem. 1995, 30(8): 767-772. DOI: 10.1016/0032-9592(95)00002-x.

Pepi, M.; Heipieper, H.J.; Fischer, J.; Ruta, M.; Volterrani, M.; Focardi, S.E. Membrane fatty acids adaptive profile in the simultaneous presence of arsenic and toluene in Bacillus sp. ORAs2 and Pseudomonas sp. ORAs5 strains. J. Extremophiles. 2008, 12(3): 343-349. DOI: 10.1007/s00792-008-0147-9.

Page 3, line 106-108:

Cell proliferation was measured by assessing the turbidity (optical density at 560 nm) of cell suspensions using a spectrophotometer, following the methodology specified in a prior study [26].

3.6  Optimization of medium ingredients

2.4.2. Plackett-Burman design (PBD)

L(127). What were the numerical values of the -1 and +1 levels?

Response: Thank you very much for your comment. This is our oversight. We forgot to indicate that it is in Supplementary Table S1, so we have made the supplement. In this study, six variables were chosen for PBD at two levels indicated by high (+1) and low (-1), referencing Supplementary Table S1 for precise values.

Page 4, line 119-121:

According to the results from the SFD, six variables, including glucose, peptone, yeast extract, KH2PO4, MgSO4, and CuSO4, were selected for PBD at two levels denoted by high (+1) and low (−1) (Supplementary Table S1).

Supplementary Table S1. Levels of the variables and statistical analysis in PBD in OD560nm for medium ingredient

Code

Variable

Level

Regression coefficient

F values

P values

Low (-1)

High (1)

X1

Glucose concentration (g/L)

200

400

0.0733

38.43

0.004

X2

Peptone concentration (g/L)

20

80

0.0150

0.83

0.418

X3

Yeast extract concentration (g/L)

10

30

-0.0483

16.59

0.016

X4

KH2PO4 concentration (g/L)

2

6

0.0550

13.29

0.024

X5

MgSO4 concentration (g/L)

0

0.8

0.0350

3.73

0.085

X6

CuSO4 concentration (g/L)

0

0.02

0.0500

4.87

0.068

3.7 2.5.2. Plackett-Burman design (PBD)

L(159). What were the numerical values of the -1 and +1 levels?

Response: Thank you very much for your comment. As the above question, this is our oversight. We forgot to indicate that it is in Supplementary Table S3, so we have made the supplement. In this study, six variables were chosen for PBD at two levels indicated by high (+1) and low (-1), referencing Supplementary Table S2 for precise values.

Page 7, line 151-153:

According to the results from the SFD, five factors including initial pH, inoculum size, rotation speed, fermentation temperature and fermentation time were selected for PBD at two levels denoted by high (+1) and low (−1) (Supplementary Table S2).

Supplementary Table S2. Levels of the variables and statistical analysis in PBD in OD560nm for fermentation condition

Code

Variable

Level

Regression coefficient

F values

P values

Low (-1)

High (1)

X1

Initial pH

5

7

4.01

3.99

0.076

X2

Inoculum size (v/v)

0.5

1.5

17.55

39.85

0.004

X3

Rotation speed (r/min)

90

150

0.1325

3.82

0.078

X4

Fermentation temperature (°C)

15

25

2.255

112.03

0.000

X5

Fermentation time (d)

1

3

8.76

76.13

0.001

  1. Results and Discussion

3.1.1. Single-factor design (SFD) of medium ingredients. L(302).

3.1.4. Response Surface Analysis (RSA) of medium ingredients. L(365).

3.2.1. Single-factor design (SFD) of fermentation conditions. L(431).

3.2.4. RSA of fermentation conditions. L(477).

It is recommended that the size of this series of graphs be increased in order to facilitate a more comprehensive appreciation of the data presented. Furthermore, it is advised that a brief description of each graph be included (a, b, c, etc.).

Response: Thank you very much for your comment. According to the comments, we adjusted the pixel density of all graphs to 600dpi to ensure that every detail in the images was accurately presented. Meanwhile, brief descriptions have been added after each graph.

Page 11-12, line 295-299:

Figure 1. Optimization of medium ingredient by SFD. (a) Types of basic culture medium, (b) Glucose concentration, (c) Peptone concentration and yeast extract concentration, (d) KH2PO4 concentration, (e) MgSO4 concentration, (f) CuSO4 concentration. The same letters in the column indicate that the data do not differ significantly at 5% probability using Tukey’s test. The blue bar means the better conditions on the growth of S. fibuligera Y1402.

Page 13, line 360-364:

Figure 2. Response surface and contour map of the interaction of various factors affecting the cell density of S. fibuligera Y1402. (a) Interaction between glucose concentration and yeast extract concentration; (b) Interaction between glucose concentration and KH2PO4 concentration; (c) Interaction between yeast extract concentration and KH2PO4 concentration. The colour change from blue to red corresponds to increase in responses.

Page 15, line 427-430:

Figure 3. Optimization of fermentation condition by SFD. (a) Initial pH and fermentation temperature, (b) Inoculum size, (c) Rotation speed and liquid volume, (d) Fermentation time. The same letters in the column indicate that the data do not differ significantly at 5% probability using Tukey’s test. The blue bar means the better conditions on the growth of S. fibuligera Y1402.

Page 16, line 474-478:

Figure 4. Response surface and contour map of the interaction of various factors affecting the cell density of S. fibuligera Y1402. (a) Interaction between inoculum size and fermentation temperature, (b) Interaction between inoculum size and fermentation time, (c) Interaction between fermentation time and fermentation temperature. The colour change from blue to red corresponds to increase in responses.

Page 17, line 502-504:

Figure 5. The growth curve of S. fibuligera Y1402 in a 3 L fermenter. The same letters in the column indicate that the data do not differ significantly at 5% probability using Tukey’s test. The blue bar means the best conditions on the growth of S. fibuligera Y1402.

  1. Please strain names in italics

Response: Thank you very much for your comment. According to the comments, we checked all strain names in italics.

      6. In the data presented as O.D. (optical density), it is recommended that these data be replaced with quantifiable values, such as CFU/mL or biomass g/L, in order to facilitate more accurate analysis of yield and productivity. Optical density provides an indication of the number of cells in a culture but lacks a unit of measurement.

Response: OD represents the optical density of absorbed light by the detected substance. Within a certain range, the concentration of the bacterial solution is linearly related to the optical density value. Therefore, we used OD values to infer the changes in yeast growth and reproduction process. Meanwhile, we consider CFU/mL or biomass g/L, as you mentioned, to be more accurate in representing the total number of cells, especially the number of viable bacteria. Therefore, we measured the CFU of yeast cells under the optimal growth medium and fermentation conditions at page 17, lines 511-512 and line 515-517, respectively. According to the results of our previous research, there is a good positive correlation between the OD value of yeast growth and its total colony count. In order to improve the efficiency of research, we use OD value to quickly reflect the optimization culture conditions. Under the best conditions, we detected the number of viable bacteria because we subsequently carried out the research on optimization of fermentation tank expansion culture conditions and preparation of active dry yeast. Initially, we used two methods: wet weight and OD value. However, due to lack of experience at that time, yeast cells were difficult to be completely centrifuged under high glucose solution conditions. Therefore, we finally only chose OD value as the indicator to quickly reflect the results. But when we optimized the fermentation tank expansion culture, we found a good method, that is, to dilute the culture first and then centrifuge and weigh it. Therefore, in the subsequent research reports, we combined OD value with wet weight for characterization, and there was also a good positive correlation between them.

    7. What was the point of planning so many experiments? The whole thing could have been reduced to a Plackett–Burman planning, and an optimization with a box Behnken design.

Response: Thank you very much for your comment. In our experiment, we conducted four experimental steps to optimize the composition of the culture medium and fermentation conditions respectively: Single-factor design (SFD), Plackett-Burman Design (PBD), Steepest Ascent Test (SAT), and Box Behnken Design (BBD). SFD served as the foundation for all optimization designs, as it was essential for determining the combinations in PBD. SAT was conducted based on the significant factors identified in the PBD for the growth of the Saccharomycopsis fibuligera Y1402. The results of PBD and SAT experiments were used to determine the most significant components and central values in the BBD experiment, resulting in the most reasonable optimized combination. Therefore, we consider the optimization of these four steps to be necessary. Actually, we learned a lesson from this. We also wanted to simplify the optimization steps through these statistical experimental methods before, but it was found that the number of factors and the range of levels we determined were not ideal, resulting in some deviation in the optimization results. Under the lesson of failure, we chose a conservative and rigorous strategy.

Reviewer 2 Report

Comments and Suggestions for Authors

The work cannot be clearly distinguished from previous studies or does not demonstrate a clear advantage over existing practices, and therefore appears redundant or of limited interest.

Scaling a fermentation process from a laboratory environment to an industrial setup presents numerous challenges that may not be evident in a 3-liter system. Factors such as heat transfer, mass transfer, and fluid dynamics change significantly as the volume of the fermenter is increased. These differences can affect the viability and efficiency of yeast cells, as well as the consistency of production.

The discussion of the components of the culture medium tested is very general, it requires being more specific in the effects at the physiological level.

Optical density measures the turbidity of a solution, which increases with the number of cells present. However, this method does not distinguish between live and dead cells. Consequently, if a significant portion of the biomass is composed of dead or damaged cells, the OD could indicate a higher level of growth than is actually effective for the purposes of the fermentation process.

Although most of what is reported in the literature focuses on the production of various enzymes, this study only considers the production of biomass, leaving aside metabolites of industrial interest. What is the importance of only increasing biomass production if there is no way to know if the metabolites of interest are in concentrations that determine their value at an industrial level?

Author Response

20240430

Re: Manuscript ID. Foods-2977575 “Optimization of High-density Fermentation Conditions for Saccharomycopsis fibuligera Y1402”

Thank you very much for providing us with helpful suggestions again. In the following, we have made modifications and replies point-by-point according to your suggestions. Revisions in the manuscript are highlighted in blue.

We look forward to hearing from you at your early convenience.

Yours sincerely,

Guangsen Fan

Address: School of Food and Health, Beijing Technology and Business University, Beijing 100048, China

Tel: +86 13811497684

Responses to the comments

  1. The work cannot be clearly distinguished from previous studies or does not demonstrate a clear advantage over existing practices, and therefore appears redundant or of limited interest.

Response: Thank you very much for your comment. In our previous work, we isolated the yeast strain Saccharomycopsis fibuligera Y1402 from the Baijiu production environment. This strain exhibited high production of flavor compounds such as 3-methylthio-1-propanol and phenylethanol, which played a significant role in imparting the unique flavor of Baijiu. Additionally, in earlier experiments, we applied this strain in tobacco fermentation experiments, significantly improving the quality of tobacco flavor (related research has not been published yet). Therefore, we believed that this yeast strain had enormous potential for practical applications. It is also not difficult to see from other researchers that the strain plays an important role in many foods. However, it is a pity that there is no research on high-density fermentation culture of the strain so far, which may limit the future industrial application of the strain to some extent. Building on these research findings, we designed this experiment with the aim of optimizing the composition of the medium and fermentation conditions for S. fibuligera Y1402, to provide support for future systematic research and industrial applications of S. fibuligera. This fills the gap in the high-density culture of the strain, and is thus of great significance and practical value.

  1. Scaling a fermentation process from a laboratory environment to an industrial setup presents numerous challenges that may not be evident in a 3-liter system. Factors such as heat transfer, mass transfer, and fluid dynamics change significantly as the volume of the fermenter is increased. These differences can affect the viability and efficiency of yeast cells, as well as the consistency of production.

Response: Thank you very much for your comment. As you said, the scaling-up of fermentation is a very complex process, not simply a matter of volume amplification. However, only after optimizing the fermentation medium and conditions at the flask and small tank levels, can we gradually proceed to fermentation and trials with larger fermentation volumes, and make fine adjustments to the parameters. It is only after these adjustments that the fermentation conditions will be applied to actual production. Therefore, flask fermentation and small-scale fermentation tank fermentation are indispensable processes in the industrial production of strains, and are the fastest and most economical strategies, providing certain data and parameters for industrial production. However, it is important to acknowledge that there are significant differences between experiments in a 3L fermenter and actual industrial-scale production. Therefore, in subsequent experiments, we will gradually increase the fermentation scale to better simulate real production scenarios and provide technical support for industrial production.

  1. The discussion of the components of the culture medium tested is very general, it requires being more specific in the effects at the physiological level.

Response: Thank you very much for your comment. We have made appropriate supplements according to your suggestions, but due to the lack of our main goal and the physiological background of the strain, we can only make some appropriate supplements.

Page 9, line 199-201:

By comparing the composition of different culture media, it can be found that the con-centration of carbon sources in medium 1 is low and the kinds of minerals and growth factors are relatively simple [32].

Page 9, line 202-205:

The iron in media 3, 4, and 5 may prevent the action of yeast Y1402-related enzymes, making them less suitable for the yeast's growth, even if they all provide a significant amount of carbon and nitrogen sources as well as an array of nutrients [5, 33].

Page 10, line 214-216:

Carbon source is the primary energy source for microbial growth and serves as the carbon skeleton for numerous substances in microbial cell metabolism. Sufficient carbon sources are crucial for microbial growth [35].

Page 10, line 219-223:

Insufficient glucose concentration below 300 g/L hindered the growth of S. fibuligera Y1402 due to limited energy supply. Conversely, excessively high glucose concentra-tions interfered with yeast's metabolism due to high osmotic pressure or the generation of numerous metabolites, such as higher alcohols, ultimately impeding its growth [36].

Page 10, line 227-230:

Yeast extract contains not only abundant nitrogen sources such as amino acids and peptides, providing necessary nitrogen sources for yeast cell protoplasm and other materials, but also various growth factors such as vitamins, trace elements, yeast nucleic acids, yeast polysaccharides and other nutrients.

Page 10, line 238-241:

  1. fibuligera Y1402 didn't grow well at lower yeast extract concentrations because it didn't have enough nitrogen sources and growth factors. High amounts of yeast extract, on the other hand, stunted its development by creating an unfavorable car-bon-to-nitrogen ratio.

Page 10, line 254-258:

Inorganic salts are crucial for the development of microorganisms, as they actively contribute to and maintain the integrity of microbial cells. They are constituents of specific enzymes and have an impact on enzyme functioning in biological processes, controlling and sustaining osmotic pressure, hydrogen ion con-centration, and redox potential during microorganism growth [21, 42].  

Page 11, line 263-270:

Phosphorus is an important component of nucleic acids, phospholipids, and coenzymes and can participate in the phosphorylation process of carbohydrate metabolism to produce high-energy phosphate compounds such as ATP to store and transport energy. It can also act as a buffer for pH in the growth environment. Therefore, a certain amount of phosphate is beneficial to the growth of yeast Y1402 [23]. Magnesium ions affect sugar conversion rates and coenzyme factor metabolism path-ways They decrease the permeability of the cytoplasmic membrane and enhance the yeast cell's resistance to metabolites such as ethanol [44-49].

Page 11, line 280-285:

Copper, being a redox-active metallic element, is indispensable in almost all living cells. It serves as a co-factor in several enzyme structures and metabolic processes, hence playing a vital part in the development and metabolism of organisms. Microorganisms may use copper to enhance the function of superoxide dis-mutase, a protective enzyme that eliminates excessive free radicals and defends the organism against oxidative damage [50, 51].

Page 11, line 289-292:

This might be attributed to the fact that high concentrations of copper ions decreased the utilization rate of glucose in yeast, inhibited energy metabolism, damaged cell membrane integrity, and affected the absorption of other metal elements by S. fibuligera Y1402 [51, 52].

  1. Optical density measures the turbidity of a solution, which increases with the number of cells present. However, this method does not distinguish between live and dead cells. Consequently, if a significant portion of the biomass is composed of dead or damaged cells, the OD could indicate a higher level of growth than is actually effective for the purposes of the fermentation process.

Response: Thank you very much for your comment. We fully agree with your viewpoint.  OD represents the optical density of absorbed light by the detected substance. Within a certain range, the concentration of the bacterial solution is linearly related to the optical density value. Therefore, we used OD values to infer the changes in yeast growth and reproduction process. Meanwhile, we consider CFU/mL or biomass g/L, as you mentioned, to be more accurate in representing the total number of cells, especially the number of viable bacteria. Therefore, we measured the CFU of yeast cells under the optimal growth medium and fermentation conditions at page *, lines 364 and page *, line 477, respectively. In addition, from our previous research on this strain, we found that there was a good positive correlation between the OD560 number of the strain and the number of viable cells measured by colony counting. Therefore, we selected the turbidity measured at this wavelength to reflect the growth of yeast, which can be supported by data on the effect of optimized culture time on strain cells growth. Otherwise, turbidity method would not be used as an important method for drawing microbial growth curves in microbiology textbooks. In order to improve the efficiency of research, we use OD value to quickly reflect the optimization culture conditions. Under the best conditions, we detected the number of viable bacteria because we subsequently carried out the research on optimization of fermentation tank expansion culture conditions and preparation of active dry yeast. Initially, we used two methods: wet weight and OD value. However, due to lack of experience at that time, yeast cells were difficult to be completely centrifuged under high glucose solution conditions. Therefore, we finally only chose OD value as the indicator to quickly reflect the results. But when we optimized the fermentation tank expansion culture, we found a good method, that is, to dilute the culture first and then centrifuge and weigh it. Therefore, in the subsequent research reports, we combined OD value with wet weight for characterization, and there was also a good positive correlation between them.

  1. Although most of what is reported in the literature focuses on the production of various enzymes, this study only considers the production of biomass, leaving aside metabolites of industrial interest. What is the importance of only increasing biomass production if there is no way to know if the metabolites of interest are in concentrations that determine their value at an industrial level?

Response: Thank you very much for your comment. Just as we stated when answering your first question, in our previous work, we isolated the yeast strain Saccharomycopsis fibuligera Y1402 from the Baijiu production environment. This strain exhibited high production of flavor compounds such as 3-methylthio-1-propanol and phenylethanol, which played a significant role in imparting the unique flavor of Baijiu. Additionally, in earlier experiments, we applied this strain in tobacco fermentation experiments, significantly improving the quality of tobacco flavor (related research has not been published yet). Therefore, we believed that this yeast strain had enormous potential for practical applications. However, it is a pity that there is no research on high-density fermentation culture of the strain so far, which may limit the future industrial application of the strain to some extent. Building on these research findings, we designed this experiment with the aim of optimizing the composition of the medium and fermentation conditions for S. fibuligera Y1402, to provide support for future systematic research and industrial applications of S. fibuligera. This fills the gap in the high-density culture of the strain, and is thus of great significance and practical value.

Of course, subsequent research will delve deeper into the metabolites (including but not limited to enzymes) and functional characteristics of S. fibuligera Y1402 in the future.

Round 2

Reviewer 1 Report

Comments and Suggestions for Authors

The new version of the article entitled “Optimization of High-density Fermentation Conditions for Saccharomycopsis fibuligera Y1402 through Response Surface Analysis” presents a significant improvement. However, some details must be adjusted;

I think it's better to put full titles in the following sections. In the text it looks cool but in titles the full name looks better.

2.4.1. SFD

2.4.2. PBD

2.4.3. SAT

2.4.4. RSA

2.5.1. SFD

2.5.2. PBD

2.5.3. SAT

2.5.4. RSA

3.1.1. SFD

The quality of figures 1, 3 is still very poor

I think the conclusion should be redone, it looks like an abstract

Thank you very much

Author Response

20240509

Re: Manuscript ID. Foods-2977575 “Optimization of High-density Fermentation Conditions for Saccharomycopsis fibuligera Y1402 through Response Surface Analysis”

Thank you very much for providing us with helpful suggestions again. In the following, we have made modifications and replies point-by-point according to your suggestions. Revisions in the manuscript are highlighted in red.

We look forward to hearing from you at your early convenience.

Yours sincerely,

Guangsen Fan

Address: School of Food and Health, Beijing Technology and Business University, Beijing 100048, China

Tel: +86 13811497684

Responses to the comments

The new version of the article entitled “Optimization of High-density Fermentation Conditions for Saccharomycopsis fibuligera Y1402 through Response Surface Analysis” presents a significant improvement. However, some details must be adjusted;

Response: Dear Reviewer, thank you very much for your valuable suggestions on our manuscript and for your recognition of our last revised manuscript. We apologize for the problems that still exist. We have made revisions one by one based on your opinions. Please review them.

  1. I think it's better to put full titles in the following sections. In the text it looks cool but in titles the full name looks better.

2.4.1. SFD

2.4.2. PBD

2.4.3. SAT

2.4.4. RSA

2.5.1. SFD

2.5.2. PBD

2.5.3. SAT

2.5.4. RSA

3.1.1. SFD

Response: Dear Reviewer, thank you very much for your advice. We felt the same way when we wrote this part. We had to deal with it reluctantly in this way because we wanted to keep it consistent with the abbreviation we used before. We agree with you completely and have revised it according to your suggestion. Thank you very much.

Page 3, line 109

2.4.1. Single-factor design of medium ingredients

Page 4, line 116

2.4.2. Plackett-Burman design of medium ingredients

Page 5, line 125

2.4.3. Steepest ascent test of medium ingredients

Page 5, line 134

2.4.4. Response surface analysis of medium ingredients

Page 6, line 144

2.5.1. Single-factor design of fermentation condition

Page 6, line 148

2.5.2. Plackett-Burman design of fermentation condition

Page 7, line 157

2.5.3. Steepest ascent test of fermentation condition

Page 8, line 163

2.5.4. Response surface analysis of fermentation condition

Page 9, line 190

3.1.1. Single-factor design of medium ingredients

Page 12, line 298

3.1.2. Plackett-Burman design of medium ingredients

Page 12, line 310

3.1.3. Steepest ascent test of medium ingredients

Page 12, line 317

3.1.4. Response surface analysis of medium ingredients

Page 14, line 365

3.2.1. Single-factor design of fermentation conditions

Page 16, line 431

3.2.2. Plackett-Burman design of fermentation conditions

Page 16, line 438

3.2.3. Steepest ascent test of fermentation conditions

Page 16, line 449

3.2.4. Response surface analysis of fermentation conditions

  1. The quality of figures 1, 3 is still very poor

Response: We are very sorry, we have rechecked and replaced it. Thank you very much.

Page 12, line 292

Figure 1. Optimization of medium ingredient by SFD. (a) Types of basic culture medium, (b) Glucose concentration, (c) Peptone concentration and yeast extract concentration, (d) KH2PO4 concentration, (e) MgSO4 concentration, (f) CuSO4 concentration. The same letters in the column indicate that the data do not differ significantly at 5% probability using Tukey’s test. The blue bar means the better conditions on the growth of S. fibuligera Y1402.

Page 16, line 425

Figure 3. Optimization of fermentation condition by SFD. (a) Initial pH and fermentation temperature, (b) Inoculum size, (c) Rotation speed and liquid volume, (d) Fermentation time. The same letters in the column indicate that the data do not differ significantly at 5% probability using Tukey’s test. The blue bar means the better conditions on the growth of S. fibuligera Y1402.

  1. I think the conclusion should be redone, it looks like an abstract

Response: Thank you very much. We have revised the conclusion part as follows:

Fermentation under the optimal conditions as an initial pH of 6, an inoculum size of 1.10 %, a rotation speed of 120 r/min, a fermentation temperature of 21 °C, a liquid volume of 12.50 mL/250 mL, and a fermentation time of 53.50 hours, with the optimal mediums as a glucose concentration of 360.61 g/L, a peptone concentration of 50 g/L, a yeast extract concentration of 14.65 g/L, KH2PO4 concentration of 5.49 g/L, MgSO4 concentration of 0.40 g/L, and a CuSO4 concentration of 0.01 g/L, the OD560nm value reached 90.11. The total colony count reached 5.50×109 CFU/mL. This study has guiding significance for the future industrial production and application of S. fibuligera. (Page 19, line 509-516)

Thank you very much again.

Reviewer 2 Report

Comments and Suggestions for Authors

The authors addressed the comments appropriately.

Author Response

20240505

Re: Manuscript ID. Foods-2977575 “Optimization of High-density Fermentation Conditions for Saccharomycopsis fibuligera Y1402 through Response Surface Analysis”

Thank you very much for providing us with helpful suggestions again and for your recognition of our last revised manuscript.

Thanks again.

Yours sincerely,

Guangsen Fan

Address: School of Food and Health, Beijing Technology and Business University, Beijing 100048, China

Tel: +86 13811497684
